# Findings from a Nationwide Study on Alcohol Consumption Patterns in an Upper Middle-Income Country

**DOI:** 10.3390/ijerph19148851

**Published:** 2022-07-21

**Authors:** Tania Gayle Robert Lourdes, Hamizatul Akmal Abd Hamid, Mohd Ruhaizie Riyadzi, Wan Shakira Rodzlan Hasani, Mohd Hatta Abdul Mutalip, Norli Abdul Jabbar, Halizah Mat Rifin, Thamil Arasu Saminathan, Hasimah Ismail, Muhammad Fadhli Mohd Yusoff

**Affiliations:** 1Institute for Public Health, National Institutes of Health, Ministry of Health Malaysia, Shah Alam 40170, Malaysia; hamizatul_ah@moh.gov.my (H.A.A.H.); mohdruhaizie@moh.gov.my (M.R.R.); wshakira@moh.gov.my (W.S.R.H.); m.hatta@moh.gov.my (M.H.A.M.); halizah.matrifin@moh.gov.my (H.M.R.); thamilarasu.s@moh.gov.my (T.A.S.); hasimah.i@moh.gov.my (H.I.); fadhli_my@moh.gov.my (M.F.M.Y.); 2Disease Control Division, Ministry of Health Malaysia, Putrajaya 62590, Malaysia; norli_aj@moh.gov.my

**Keywords:** alcohol drinkers, above 15 years, NHMS, Malaysia

## Abstract

Alcohol consumption is a risk factor for various diseases, especially non-communicable diseases (NCDs) and injuries. The reduction of the harmful use of alcohol is mentioned in Target 3.5 of the Sustainable Developmental Goals (SDG). This study aimed to determine factors associated with current alcohol drinking among Malaysians aged 15 years and above. Data from the National Health and Morbidity Survey (NHMS) 2019, a cross-sectional nationwide survey using a two-stage stratified random sampling design, was used. Current alcohol drinking was defined as having consumed any alcoholic beverage in the past 12 months. Descriptive statistics and multiple logistic regression analysis were employed. The prevalence of current drinkers was 11.5%; 95% CI: 9.8, 13.5. The majority (65%) consumed alcoholic beverages in Category 2, which is mainly beer. Current drinkers consumed alcohol mostly once a month (59.0%), one or two drinks (71.1%), on a typical day. Most respondents had six or more drinks less than once a month (27.6%). Sex, age, ethnicity, education, employment, and smoking were found to be significantly associated with current alcohol drinking. Marital status, locality of residence, and household income were not significantly associated. Alcohol drinking is a problem among certain vulnerable groups and should be tackled appropriately.

## 1. Introduction

The harmful use of alcohol, which includes binge drinking and heavy episodic drinking (HED), not only causes health and social consequences for the drinker but also for the family and society at large. The Sustainable Developmental Goals (SDGs) in health Target 3.5 (“Strengthen the prevention and treatment of substance abuse, including narcotic drug abuse and harmful use of alcohol”) particularly emphasize the harmful use of alcohol [1]. Alcohol consumption is undoubtedly a risk factor for non-communicable diseases (NCDs) [2]. The Global Action Plan for the Prevention and Control of Non-communicable Diseases (2013–2020) reported that the harmful use of alcohol is one of the four behavioral risk factors (tobacco use, unhealthy diet, physical inactivity, alcohol use) for three major NCDs (cardiovascular disease, cancer, chronic respiratory disease) [3].

A vast number of other diseases and injuries are also attributed to drinking alcohol, including alcohol dependence, liver cirrhosis, injuries due to violence, suicides, and road traffic accidents [1]. The attributable disability-adjusted life years (DALYs) were highest for tobacco smoking (170.9 million DALYs), followed by alcohol (85.0 million), and illicit drugs (27.8 million) [4]. According to a recent meta-analysis, there was evidence that a reduction of alcohol consumption among those who drank more than two drinks a day significantly reduced their blood pressure [5].

Several socio-demographic factors have an influence on alcohol consumption since it is very much related to social culture. Males and smokers were more likely to consume alcohol according to findings from a study done among university students in Myanmar [6]. This group was also more likely to binge drink [7]. A couple of studies found that age, ethnicity, gender, education levels, income, employment, and locality of residence were significantly associated with current drinking [8,9].

With an expected overall growth of 176 percent from 2000 to 2019, Asia has the fastest-growing alcohol market, accounting for nearly 30 percent of worldwide alcohol sales in 2014. China and India are leading the rise with rates of 382 percent and 1245 percent, respectively [10]. Malaysia, along with Indonesia which is also a Muslim majority country, undoubtedly have a low global average for alcohol consumption [11]. However, there are certain ethnicities such as Chinese and Bumiputera Sabah and Sarawakians [12,13,14] who have rather higher consumptions of alcohol and risky drinking patterns compared to other ethnicities in Malaysia. A local study among millennials found that binge drinking posed an increased association with a number of behavioral risk factors and harmful alcohol-related outcomes [7].

In Malaysia, alcohol consumption reduction is targeted on its harmful use and is not aimed at any particular community. Therefore, this paper aims to showcase and strengthen current evidence on the factors associated with alcohol drinking in Malaysia, which includes socio-demographic characteristics of current drinkers, types of alcoholic beverages consumed, and frequencies of drinking.

## 2. Materials and Methods

Data from the National Health and Morbidity Survey (NHMS) 2019 were used in this study. The NHMS is the largest nationwide health survey conducted in Malaysia by the Institute for Public Health. This study, which examines a variety of community health-related subjects, is undertaken on a regular basis. In the NHMS, a two-stage stratified random sampling was used to ensure a nationally representative sample in terms of geography, socio-demographics, and economic diversity in Malaysia. The first stage of sampling was the Enumeration Blocks (EBs), which formed the sampling frame. The second stage of sampling was the Living Quarters (LQs). The selection of EBs was done within each state (primary stratum) and within urban or rural areas (secondary stratum) with assistance from the Department of Statistics Malaysia (DOSM). Malaysia consists of 13 states and 3 federal territories. For this study, a total of 475 EBs were selected and 5676 LQs were obtained from those EBs (12 LQs made up 1 EB). A total of 362 and 113 EBs were selected from urban and rural areas, respectively [9]. Data collection was conducted between July to October 2019. The respondents who answered the Alcohol Consumption module in NHMS 2019 were all 13 years of age and above. However, for this study, sub-analysis for those 15 years old and above (n = 11,111) was done to suit international definitions and standards [1].

### 2.1. Tools

The study’s instrument was a self-administered standardized Adult Use Disorders Identification Test (AUDIT) questionnaire. The AUDIT is a 10-item tool developed by the World Health Organization (WHO) that assesses alcohol consumption, drinking behavior, and alcohol related problems [15,16]. This questionnaire was made available in four languages to suit the local people: English, Bahasa Malaysia, Tamil, and Mandarin. Two screening questions on ever and current drinking were included before answering the AUDIT. Data on the respondents’ socio-demographic details, such as age, income, marital status, level of education, and locality of residence, were also collected. In the AUDIT, each question has a set of possible responses, with a score ranging from 0–4 (items 1 to 8 are scored on a 0–4 scale and items 9 and 10 are scored 0, 2, 4). A cut-off score of 8 or more indicates a hazardous or harmful pattern of drinking [14]. The questionnaire is further categorized into three domains: Items 1, 2, and 3 (hazardous alcohol use), Items 4, 5, and 6 (dependence symptoms), and Items 7, 8, 9 and 10 (harmful alcohol use). Item 1 (frequency of drinking); Item 2 (typical quantity); Item 3 (frequency of heavy drinking); Item 4 (impaired control over drinking); Item 5 (increased salience of drinking); Item 6 (morning drinking); Item 7 (guilt after drinking); Item 8 (blackouts); Item 9 (alcohol-related injuries); Item 10 (others concerned about drinking). A written informed consent and an assent for minors (less than 18 years old) was taken prior to the interviews from parents or guardians. This questionnaire was self-administered by the respondents to encourage truthfulness and confidentiality.

### 2.2. Definition of Alcohol Drinkers

Those who had drunk any alcoholic beverage in the previous 12 months were considered current drinkers. Ever drinkers are those who have drunk alcohol at least once in their lives. Six or more alcoholic standard drinks consumed in one sitting was considered binge drinking. HED was defined as six or more standard alcoholic beverages consumed in a single session at least once per week [17]. Lifetime abstainers were defined as those who have never consumed alcohol. Abstainers in the past 12 months were defined as those who did not drink any alcohol in the past 12 months. Types of alcoholic beverages were grouped into 5 categories according to their alcohol content; Category 1 (alcohol content < 2%): shandy; Category 2 (alcohol content < 9%): beer including lager, ale or stout; Category 3 (alcohol content 10–25%): wine, cider, champagne, *Peri*, toddy, *Tuak*, *Tuak Kelapa*, *Bahar, Lihing or Ijok;* Category 4 (alcohol content > 30%): brandy, rum, whisky, vodka, *gin, Samsu, Samsu Cheng, Montoku* or *Langkau*, and others.

### 2.3. Covariates

A total of nine covariates were included in this study (sex (male/female), age (15 to 19 years/20 to 39 years/40 to 59 years/60 years and above), location (rural/urban), ethnicity (Malay/Chinese/Indian/Bumiputera Sabah and Sarawak/Others), marital status (single, divorcee, widow/widower, married), education level (no formal education/primary education/secondary education/tertiary education), income (bottom 40%/ middle 40%/top 20%), employment status (employed/unemployed, retiree, homemaker, caregiver), and current smoking (yes/no). Current smokers were defined as those who were currently using any smoked tobacco products (manufactured cigarettes, hand-rolled cigarettes, kretek, cigars, shisha, bidis, or tobacco pipes).

### 2.4. Statistical Analysis

Descriptive statistics were used to present data on types of alcoholic beverages consumed and the frequencies of each item in the AUDIT questionnaire. Multiple logistic regression analysis was employed to determine the factors associated with alcohol consumption among Malaysians aged 15 years and above. All possible two-way interactions between the independent variables were assessed in producing the final model. The fit of the model was examined using a classification table. Data were presented with a 95% confidence level. All statistical analyses were carried out using statistical software SPSS version 26 (IBM Corporation, Armonk, NY, USA) with complex sample function.

## 3. Results

### 3.1. Socio-Demographic Characteristics of Current Drinkers in Malaysia

The overall response rate of this survey was 87.2% [12]. The mean age of the population in this study was 40.8 (SD ± 15.63). The prevalence of current drinkers among Malaysians aged 15 years and above was 11.5%; 95% CI: 9.8, 13.5. Males (16.3%; 95% CI: 13.9, 19.1) had a higher prevalence compared with females (6.4%; 95% CI: 5.0, 8.2). Those in age groups 20 to 39 years (13.4%; 95% CI: 10.9, 16.3) and 40 to 59 years (12.2%; 95% CI: 10.1, 14.7) had a higher prevalence of current drinkers compared with those in other age groups (Table 1).

Urban areas saw a prevalence of current drinkers of 11.8% (95% CI: 9.8, 14.2) compared with rural areas that had a 10.4% (95% CI: 7.9, 13.6) prevalence.

Current drinking was more prevalent among those employed (15.2%; 95% CI: 12.9, 17.8) compared with the unemployed (5.8%; 95% CI: 4.3, 7.8). According to marital status, those who were single, may it be divorced or widowed, had a prevalence of 12.3% (95% CI: 9.8, 15.4) current drinkers compared with those who were married (10.9%; 95% CI: 9.2, 12.9). The ethnic subgroup that had the highest prevalence was the Bumiputera of Sabah and Sarawak (31.1%; 95% CI: 24.3, 38.8). Those with a household income in the top 20% (17.1%; 95% CI: 12.6, 22.7) category had a higher prevalence of current drinkers compared with the other income categories. Current smokers (19.2%; 95% CI: 15.4, 23.6) had a higher prevalence of alcohol drinkers compared with non-current smokers (9.5%; 95% CI: 7.9, 11.4) (Table 1).

### 3.2. Types of Alcoholic Beverages and Frequency

Figure 1 shows the preference of alcoholic beverages among current drinkers in Malaysia. The majority (65%) consumed alcoholic beverages in Category 2, which is mainly beer. This is followed by Category 3 (17.5%), which comprises drinks like wine, champagne, and local drinks such as *tuak* and *bahaar*. Category 4 (brandy, rum, whisky, vodka, *gin, Samsu, Samsu Cheng, Montoku,* or *Langkau)*, Category 1 (shandy), and others were 7.7%, 4.1%, and 5.6% respectively. Current drinkers consumed alcohol mostly once a month (59.0%), followed by two to four times a month (23.1%) (Figure 2). Current drinkers mostly consumed one or two drinks (71.1%) on a typical day (Figure 3). Most respondents had six or more drinks less than once a month (27.6%), followed by monthly (12.3%), weekly (7.1%), and daily or almost daily (1.8%) (Figure 4).

### 3.3. Multiple Logistic Regression Analysis

Multiple logistic regression analysis (Table 2), adjusted for possible confounders, showed that males (aOR: 2.21; 95% CI: 1.65, 2.97) were 2.21 times more likely to consume alcohol compared with their female counterparts. Compared with those aged 60 years and above, all ages less than the reference group had significantly higher odds of being associated with alcohol drinking and these odds reduced for the subsequent age groups: 15–19 years (aOR: 3.23; 95% CI: 1.58, 6.58), 20–39 years (aOR: 2.04; 95% CI: 1.30, 3.20), and 40–59 years (aOR: 1.60; 95% CI: 1.02, 2.52). Considering the educational levels of the respondents, the odds of being associated with alcohol drinking was 2.28 times higher among tertiary educated respondents compared with those with no formal education. Being employed had the odds of 1.49 times of being associated with alcohol consumption compared with those unemployed. With regards to the various ethnicities of the respondents, Chinese (aOR: 4.80; 95% CI: 2.51, 9.17), Bumiputera of Sabah and Sarawak (aOR: 4.71; 95% CI: 2.46, 9.03), and Indian (aOR: 2.52; 95% CI: 1.26, 5.05) ethnics were significantly associated with alcohol drinking compared with the reference (other ethnicities). Current smokers (aOR: 2.17; 95% CI: 1.49, 3.16) were 2.17 times more likely to consume alcohol compared with non-smokers. However, marital status, locality of residence, and household income did not have a significant association with alcohol drinking. Detailed results are tabulated in Table 2.

## 4. Discussion

From this study, we found that males, those aged less than 60 years, tertiary educated, employed, ethnicity (Chinese, Bumiputera Sabah and Sarawak, Indian), and smokers were more likely to be associated with alcohol consumption. In a study conducted by Delker E, young adults were found to be particularly vulnerable to high-risk drinking and injury caused by drinking [18]. In relation to this, the evidence from this study is worrying as the results tell us that the younger age is associated with alcohol drinking. Several other studies also support this finding. For example, a study conducted in Sri Lanka reported similar age groups to be significant alcohol users and were also related to abuse and dependence [19]. Problematic alcohol consumption among male adults in Nepal was significantly associated with the 25 to 44 years’ age group [20].

With reference to gender, not only more males consumed alcohol [21,22,23] but also were more likely to be risky drinkers compared with females [14,18,24,25]. Similarly, males in this study were more likely to be current drinkers compared with females. Worryingly, despite the prevalence of current drinkers being on the lower side, almost half of them did binge drink. Although we did not analyze male binge drinking in this paper, findings from NHMS 2019 reported that 49.5% male current drinkers aged 18 years and above did binge drink [12].

In a study conducted in five states in India, having a higher income and living in urban areas were significantly associated with current drinking [8]. A higher level of household income was also reported to be positively associated with high-risk drinking among middle-aged Korean men [21,26]. Our study’s findings indeed show that a lower income is a protective factor for alcohol consumption compared with a higher income.

Our study indicated that current smokers were very likely to also be current drinkers. The international literature also support these findings, where smoking and alcohol had a mutual effect on each other’s influence of increased craving [27]. Smokers were also at an increased odds of drinking alcohol irrespective of gender [23,25,28]. Additionally, smoking and drinking concurrently also elevated the risk of hypertension [29,30]. Another study found that the proportion of smokers among drinkers was high [30], which was also consistent with our study findings where the prevalence of smokers among current drinkers was 19.2%.

The differences in alcohol consumption among communities is closely related to religion, alcohol related policies, and economic growth [31]. Malaysia, compared with other South-East Asian countries, has a low alcohol-per-capita consumption (APC). Moreover, between the years 2010 to 2017, Malaysia had a decrease in APC [11]. However, an article published in the Lancet reported global APC consumption increased from 5.9 L to 6.5 L and it would reach 7.6 L by the year 2030, and current policies are not enough to achieve a reduction of the harmful use of alcohol [31]. Religious beliefs, however, do play a part in alcohol consumption [25]. Since Malaysia is a Muslim majority country, alcohol consumption on a whole is generally low. In a study conducted by Mutalip et al., in Malaysia, ethnicity (Bumiputera Sabah and Sarawak) had the highest odds and was significantly associated with risky drinking [14]. Drinking alcohol is commonly associated with ethnic backgrounds and culture [32] and this is particularly related to the culture of the indigenous people in East Malaysia (Sabah and Sarawak), especially during festive occasions. This group was also found to be more prone to polysubstance use as reported in a study among Malaysian adolescents [33]. Interestingly, a study done in Sabah found that *Montoku* was mostly drunk by alcohol drinkers there and it was very cheap (RM 0.50 per standard drink), which boils down to the issue of taxation of locally distilled alcohol [34]. Thus, the prevention of issues related to alcohol consumption should be targeted more to people of specific ethnicities and cultures where different types of beverages are consumed.

Policies related to alcohol are well in place in Malaysia. The maximum legal blood alcohol concentration (BAC) when driving is set at 0.08% compared with 0.05% and no BAC limit in Thailand and Indonesia, respectively. The minimum age to purchase alcoholic beverages is set at 21 years old in Malaysia; whereas, Singapore and Thailand have lower legal ages of 18 years old and 20 years old, respectively [1]. Alcoholic beverages are actually quite easily accessible in Malaysia, as it is sold in neighbourhood convenience stores. A meta-analysis has shown that reducing the physical availability of take-away alcohol will reduce the APC consumption, thus reducing the relevant harm it could pose [35].

The type of alcoholic beverages preferred by Malaysian adults found in our study was from Category 2, which is mainly beer. Comparatively, current drinkers in Thailand and India preferred spirits, which are mainly brandy, rum, whiskey, and vodka [8,11]. However, a study among Chinese college students showed beer to be the preferred beverage [24].

### Strengths and Limitations

This study has several strengths and limitations. The size of the sample and the sample’s representation of the Malaysian population are its greatest assets. To our knowledge, this is the first study to disseminate updated findings related to alcohol consumption patterns in Malaysia of this scale. One of its limitations is the study design used, cross-sectional, where cause and effect could not be determined. This study design, though, was the most suitable given the large sample size and existing resources. In addition, alcohol surveys may not provide reliable per capita consumption data [36]. Secondly, the majority of the respondents in this study were Malay (Muslim). This could have caused some degree of dilution of the actual association, because alcohol drinking is strictly prohibited in Islam. Although the reduction of the harmful use of alcohol is one of the SDGs, we unfortunately could not analyze the factors associated with HED and binge drinking using this sample due to the low sample from this group. Therefore, additional research utilizing various study designs and concentrating on groups that are very likely to consume alcohol in a dangerous manner is necessary. Other factors, such as parental and peer drinking that were not explored in this study, should be included in future nationwide surveys. In the Global Status Report on Alcohol and Health 2018, the three strategies that are effective in reducing the harmful use of alcohol are undoubtedly to increase taxes on alcoholic beverages, bans or restrictions on alcohol advertising in the media, and the restriction on the availability of retailed alcohol [1,37]. To successfully reduce the harm caused by alcohol consumption, governments should work to adopt this approach.

## 5. Conclusions

This paper highlights the factors associated with alcohol consumption and the pattern of alcohol consumption in Malaysia. Although the prevalence of alcohol consumption is generally lower compared with other risk factors reported in NHMS 2019, alcohol drinking may be a problem among certain vulnerable groups. More policies directed towards this subgroup should be formulated to reduce the effects from the harmful use of alcohol.

## Figures and Tables

**Figure 1 ijerph-19-08851-f001:**
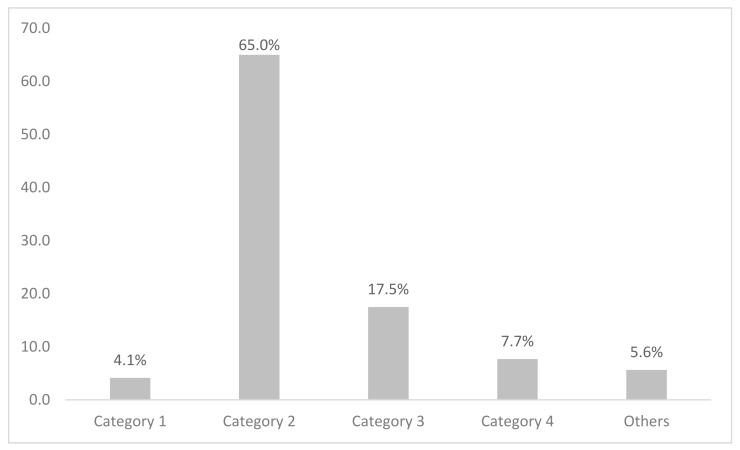
Preference of alcoholic beverages among current drinkers in Malaysia.

**Figure 2 ijerph-19-08851-f002:**
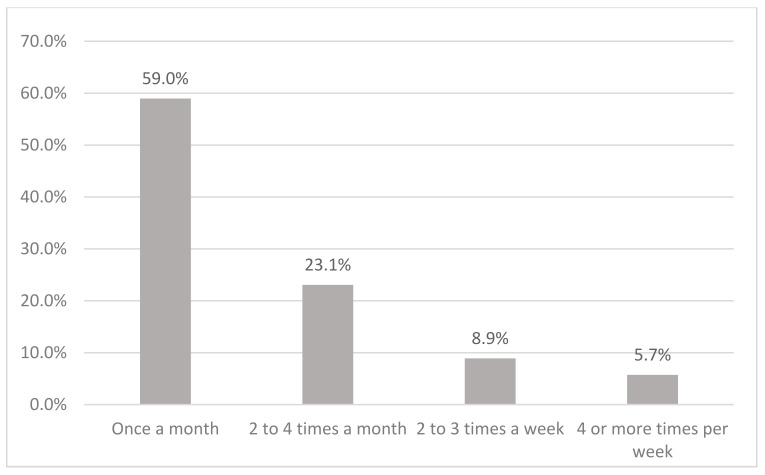
Drinking frequency among current drinkers in Malaysia.

**Figure 3 ijerph-19-08851-f003:**
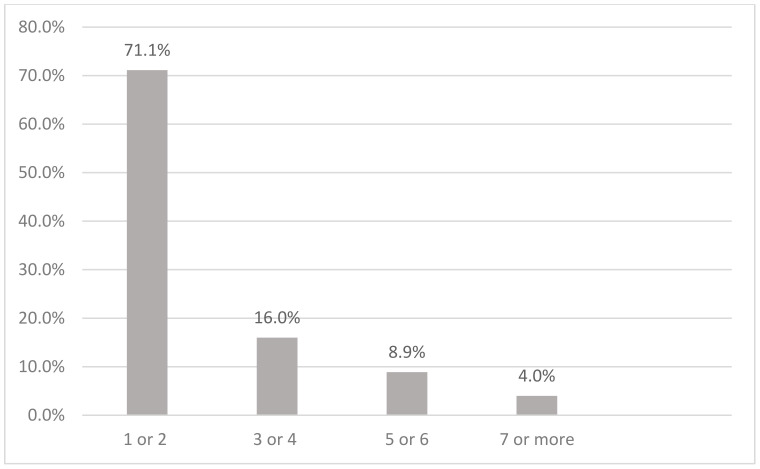
Number of drinks consumed on a typical day among current drinkers in Malaysia.

**Figure 4 ijerph-19-08851-f004:**
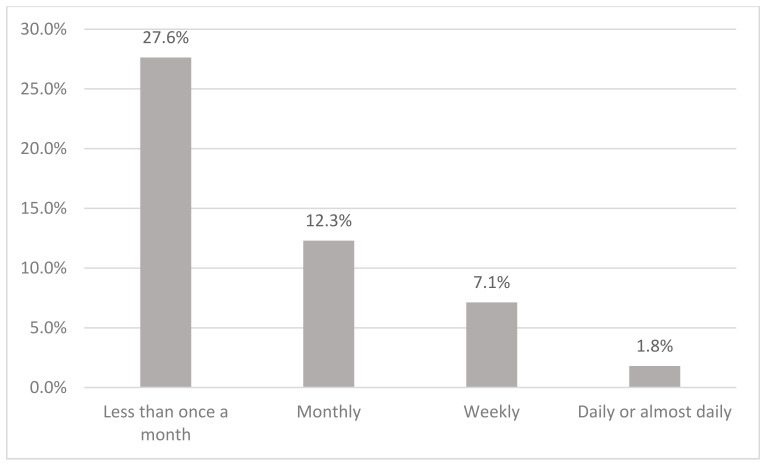
Frequency of having six or more drinks on one occasion among current drinkers in Malaysia.

**Table 1 ijerph-19-08851-t001:** Socio-demographic characteristics of current drinkers in Malaysia aged 15 and above.

Socio-Demographic Characteristics	n	Estimated Population	Prevalence	95% CI	
				LCL	UCL
Overall	818	2,640,940	11.5	9.8	13.5
Sex					
Male	576	1,923,105	16.3	13.9	19.1
Female	242	717,835	6.4	5.0	8.2
Age group					
15–19	63	221,120	8.3	5.8	11.8
20–39	359	1,453,241	13.4	10.9	16.3
40–59	279	773,470	12.2	10.1	14.7
60 years and above	117	193,110	6.2	4.8	8.0
Mean age (40.8 SD ± 15.63)					
Locality					
Rural	271	533,346	10.4	7.9	13.6
Urban	547	2,107,594	11.8	9.8	14.2
Education level					
No formal education	37	69,818	6.4	4.2	9.8
Primary education	166	558,020	12.2	9.2	16.2
Secondary education	395	1,243,281	10.7	8.8	13.1
Tertiary education	217	767,026	13.7	10.9	17.2
Occupation					
Employed	587	2,052,715	15.2	12.9	17.8
Unemployed/retiree/homemaker/caregiver	112	255,202	5.8	4.3	7.8
Marital status					
Single/divorcee/widow/widower	306	1,146,244	12.3	9.8	15.4
Married	512	1,494,696	10.9	9.2	12.9
Ethnicity					
Chinese	312	1,248,468	26.0	21.6	30.9
Malay	37	81,572	0.7	0.5	1.1
Indian	117	233,429	17.3	13.5	21.9
Bumiputera of Sabah Sarawak	298	794,696	31.1	24.3	38.8
Others	54	282,776	11.7	6.9	19.2
Household Income					
Bottom 40%	498	1,503,522	10.7	9.0	12.6
Middle 40%	184	740,137	13.3	9.8	17.8
Top 20%	105	331,450	17.1	12.6	22.7
Current smoker					
Yes	252	931,704	19.2	15.4	23.6
No	565	1,706,103	9.5	7.9	11.4

**Table 2 ijerph-19-08851-t002:** Factors associated with alcohol drinking among adults aged 15 years and above in Malaysia.

Variables	Unadjusted OR				Adjusted OR			
	Exp (B)	95% CI		*p*-Value	Exp (B)	95% CI		*p*-Value
		LCL	UCL			LCL	UCL	
Sex								
Male	2.85	2.22	3.66	<0.001	2.21	1.65	2.97	<0.001 *
Female	Ref				Ref			
Age group								
15–19	1.36	0.91	2.03	0.131	3.23	1.58	6.58	0.001 *
20–39	2.33	1.76	3.08	<0.001	2.04	1.30	3.20	0.002 *
40–59	2.09	1.50	2.92	<0.001	1.60	1.02	2.52	0.042 *
60 years and above	Ref				Ref			
Locality								
Rural	0.87	0.59	1.26	0.450	1.22	0.70	2.12	0.488
Urban	Ref				Ref			
Education level								
No formal education	Ref				Ref			
Primary education	2.04	1.18	3.51	0.010	1.96	1.00	3.83	0.049 *
Secondary education	1.75	1.07	2.87	0.026	1.83	1.02	3.28	0.044 *
Tertiary education	2.32	1.39	3.89	0.001	2.28	1.14	4.56	0.019 *
Occupation								
Employed	2.94	2.24	3.86	<0.001	1.49	1.01	2.19	0.045 *
Unemployed/retiree/homemaker/caregiver	Ref				Ref			
Marital status								
Single/divorcee/widow/widower	1.15	0.88	1.49	0.302	1.15	0.78	1.71	0.486
Married	Ref				Ref			
Ethnicity								
Chinese	2.65	1.41	4.98	0.002	4.80	2.51	9.17	<0.001 *
Malay	0.05	0.02	0.11	<0.001	0.06	0.03	0.14	<0.001 *
Indian	1.58	0.81	3.07	0.179	2.52	1.26	5.05	0.009 *
Bumiputera of Sabah Sarawak	3.41	1.75	6.66	<0.001	4.71	2.46	9.03	<0.001 *
Others	Ref				Ref			
Household Income								
Bottom 40%	0.58	0.40	0.84	0.004	0.71	0.44	1.16	0.174
Middle 40%	0.74	0.46	1.19	0.218	0.70	0.41	1.19	0.185
Top 20%	Ref				Ref			
Current smokers								
Yes	2.26	1.70	3.02	<0.001	2.17	1.49	3.16	<0.001 *
No	Ref				Ref			

* *p* value significant at <0.05; Multicollinearity and interactions were checked and not found. Classification table (overall correctly classified percentage = 87.3%) was accepted to check model fitness.

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
