# Peer review of "Findings from a Nationwide Study on Alcohol Consumption Patterns in an Upper Middle-Income Country"

_ijerph, 2022, doi:10.3390/ijerph19148851_

Round 1

Reviewer 1 Report

Comments to authors:

A fine study and analysis but I think a few things could be improved, and I believe you left out a crucial regression analysis (predicting problem drinking specifically, rather than just normal drinking). Comments to follow here:

P 5, fig 1 and commentary – should you also define categories 1 and 4 for the reader?

Fig 2 – the 3rd category should be 2-3 times per week (not per month)

Fig 3 – would it make sense to compile last to categories, so 7 or more?

P 8 line 195 – you say your results show more prevalence among youth, but this is not the case since the p value shows no statistical significance, with the middle age scores being much more associated with drinking….

P 7 chart – you argue that income has no association with alcohol use. But the reference category was the highest income, and relative to this, the lowest income is only 58% as likely to drink, at a strong level of significance. Might be better to treat lower income or middle income as the reference category, or at least change the narration to admit that low income protects against alcohol use vs high income (middle income is indeed statistically insignificant).

I think the burning question most has is predictors of “problem” drinking….. what are the best predictors of people who drink 7 or more drinks daily, for example?  As I think most people do not consider people drinking 1 drink per month as a problem worth solving. Thus I would have liked to see a final regression analysis that just treat 7 or more drinks per day as the dependent variable. That would be very interesting and easy to calculate and add into your final results and analysis.

Reviewer 2 Report

Dear authors,

I am grateful for the opportunity to review this research article. This article gives an overview of alcohol consumption patterns in the middle-income country Malaysia using data from 2019. The manuscript provides a solid overview but needs a considerable revision of the content and language. My main concerns relate to the way the results are presented. A stringent story line is missing, starting in the beginning by explaining the relevance of this work and answering the research question in the discussion. The individual sections are currently juxtaposed and the findings are not described in an interesting way. The manuscript may benefit from further more interesting analyses. Please find my detailed comments below.

I wish the authors good luck with their submission!

Introduction:

·      The introduction is very broad and general. I would love to read more information specific to Southeast Asia or Malaysia. Why is it important to look at alcohol consumption patterns in Malaysia? Instead of calling the global figures of alcohol use and harm, please consider to be more specific to the target country. Also, the paragraph about sociodemographic variations in alcohol consumption patterns comes very short and is not yet well embedded in the introduction.

·      “The harmful use of alcohol, which includes binge drinking and heavy episodic drinking (HED) not only causes health and social consequences for the drinker but also to the family and the society at large.” If you name both binge drinking and heavy episodic drinking, you need to explain the difference in both terms. In fact, both patterns describe a more or less similar drinking pattern, that is the consumption of high quantities of alcohol within a given time. It might be more interesting to specify that harmful use of alcohol includes the chronic use as well as heavy episodic drinking.

Methods:

·      To the best of my knowledge, different AUDIT thresholds are used to define harmful alcohol use and a score of 8 appears to be rather high. A reference for the use of this cut-off would be helpful.

·      I guess the difference between binge drinking and HED is that the former refers to at least one drinking occasion with 6+ drinks in the past year while the latter refers to such drinking occasions at a weekly basis? I think this definition of HED is quite problematic and not in line with those of WHO, for example (see Global Status Report 2018: https://www.who.int/substance_abuse/publications/global_alcohol_report/en/). The authors may consider to just use the term HED but specifying the reference period, for example: HED at least once a year vs. weekly HED.

·      I think it would be very interesting to explore sociodemographic variations not only in current alcohol users but also in harmful alcohol users (compared to alcohol users not surpassing the harmful alcohol use threshold) as well as in alcohol users reporting HED.

·      If possible, please indicate the survey response rate. Please also indicate how missing data was handled.

Results:

·      It is not clear whether the sociodemographic patterns indicated in section 3.1 based on the findings of the multiple regression model or not. If this is case, coefficients and significance levels need to be reported. If these results do not rely on regression analyses, they should not be described in such a way, given the fact that it is not clear whether a difference is relevant (= significant) or not. For example, is the prevalence of alcohol use in urban areas (11.8%) really higher than in rural areas (10.4%)? The 95% confidence intervals are largely overlapping… So please clarify if these findings are descriptive or based on inference statistic.

·      “Most respondents had six or more drinks less than once a month (27.6%), followed by monthly (12.3%), weekly (7.1%) and daily or almost daily (1.8%). Overall, 48.8% of current drinkers did binge drink (Figure 4).” This is not correct. Most or half of the surveyed alcohol users do not engage in HED (100 – 48.8% = 51.2%).

·      Using “others” as reference category for ethnicity might be less useful as it is unclear which ethnicities are included in this category. Therefore, the result loses significance. Another reference might be more appropriate.

Discussion:

·      In general, the discussion should begin with a brief summary of study’s the key findings.

·      “However, our study findings contradict this as factors such as household income and the locality of the residence were not associated with alcohol drinking.” It’s not necessarily a contradiction if you cannot find a previously reported association. There could be other explanations why the link is missing (also related to statistical methods).

·      The reference to religion in the discussion section appears somewhat out of place, as religion was not investigated in this study.

·      Important key limiations of alcohol surveys are missing, such as low alcohol consumption coverage, reporting bias, selective bias in sampling (e.g. doi: 10.15288/jsad.2015.76.158, doi: 10.1093/alcalc/agaa048, doi: 10.1111/dar.13148). Also, unrecorded alcohol use was not caputed but may account for a substantial proportion in this country (see other countries in SEAS: doi: 10.1111/add.14173)

Round 2

Reviewer 1 Report

On page 5, you explain the categories for 2&3, but not 1&4 -- your response was that these were previously defined in methods, which is true. But you should either define each category as a reminder to the reader, or not define any. You should not just define half of them as this is inconsistent. 

p7 -- the regression analysis should a strong predictor that ages 20-39 are more likely to drink, and to a lesser extend 40-59, in comparison to the reference category of 60+. There is no statistically significant relation to youth aged 15-19 in comparison with reference group (P far > .05). 

In the limitations section, there is a very brief mention of the need to study problem drinkers as a group. Yet it should be explained why this was not possible in the current study, and how a future study would be able to focus more on this most pressing issue. 
